# Physical theory for near-bed turbulent particle-suspension capacity.

Joris T. Eggenhuisen[1], Matthieu J.B. Cartigny[2], Jan de Leeuw[1].

[1]Department of Earth Sciences, Utrecht University, Heidelberglaan 2, 3584 CS Utrecht, the Netherlands.
[2]Department of Geography and Earth Sciences, Durham University, South Road, Durham, DH1 3LE, UK.

*Correspondence to*: Joris T. Eggenhuisen (j.t.eggenhuisen@uu.nl)

**Abstract.** The inability to capture physics of solid-particle suspension in turbulent fluids in simple formulas is holding back application of multiphase fluid dynamics techniques to many practical problems involving particle suspension in nature and society. We present a force-balance approach to particle suspension in the region near no-slip frictional boundaries of turbulent flows. The force balance parameter $\Gamma$ contains gravity and buoyancy acting on the sediment, and vertical turbulent fluid forces;

it includes universal turbulent flow scales and material properties of the fluid and particles only. Comparison to measurements shows that $\Gamma = 1$ gives the upper limit of observed suspended particle concentrations in a broad range of flume experiments and field settings. The condition of $\Gamma > 1$ coincides with complete suppression of coherent turbulent structures near the boundary in Direct Numerical Simulations of sediment-laden turbulent flow. $\Gamma$ thus captures the maximum amount of sediment that can be contained in suspension at the base of turbulent flow, and can be regarded as a suspension capacity parameter. It

can be applied as a simple concentration boundary condition in modelling studies of dispersion of particulates in environmental and man-made flows.

**Keywords:** Suspension capacity; turbulence; near-bed concentration; suspended sediment transport.

## 1 Introduction

Suspension of solid particles in turbulent fluid flow is one of the most widely occurring physical phenomena in nature, yet no physical theory predicts the particle suspension capacity of the wind, avalanches, pyroclastic flows, rivers, and estuarine or marine currents. Classic diffusion solutions for the distribution of suspended particles within turbulent flows (McTigue, 1981; Montes Videla, 1973; Rouse, 1937; Vanoni, 1940) do not predict the absolute particle concentration at any one location in the flow, but describe the relative concentration with respect to the concentration $C_b$ at a variably, often poorly, defined reference

elevation some millimetres or centimetres above the boundary for typical river scales [see García ( 2008) for an overview]. The absolute sediment transport capacity of turbulent flows can therefore not be determined with a diffusion approach, and is in need of closure. Attempts to deduce capacity to suspend particles from turbulent stresses and buoyancy considerations (Bagnold, 1966; Leeder, 1983, 2007; Leeder et al., 2005) have not solved the closure problem, and empiric formulations for near-boundary sediment suspension (Garcia and Parker, 1991, 1993; van Rijn, 1984; Smith and McLean, 1977; Zyserman and

Fredsoe, 1994) have been widely used to close simulations of the particle concentration field. The aim of this paper is to establish a simple formula that captures the dynamics of turbulent particle suspension. The objective is a relation between sediment concentration, flow conditions, and material properties, which can be easily parameterised. This will allow inclusion of the physical control on turbulent sediment suspension in field studies when solution of the full equations governing multiphase turbulent transport is outside of the scope (e.g. Kane et al., 2016). In order to achieve this objective we establish a

force-balance parameter $\Gamma$ that compares turbulent forces near the boundary of a turbulent suspension to gravity and buoyancy forces acting on suspended particles. Comparison with data indicates that the theory predicts the absolute value of the near-boundary reference concentration $C_b$ for $\Gamma = 1$. Additionally, it is shown that values of $\Gamma < 1$ lead to turbulence extinction over a wide variety of sediment concentrations as observed in the Direct Numerical Simulations of Cantero et al. (2011) and the grid-box experiments of Bennett et al. (2014). Thus there is now a consistent framework of experiments, modelling, and theory

that establishes that turbulence extinction due to saturation with suspended particles is not, as commonly perceived, a high-concentration effect, but occurs at a sediment concentration that is set by flow conditions and material properties of the particles and supporting turbulent fluid. Resolving the capacity of turbulent flows to suspend particles with a concentration of $C_b$ near the bed will allow determination of the sediment transport capacity of the entire flow when combined with classic turbulent diffusion approaches.

## 2 Theory for suspension capacity

### 2.1 Derivation

Suspended particles of density $\rho_s$ with a volumetric sediment concentration $C_b$ have a weight $W$ per unit volume equal to

$$W = C_b \rho_s g , \qquad (1)$$

and experience an upward directed buoyancy force $F_b$ equal to the weight of fluid displaced by particles

$$F_b = C_b \rho_f g . \qquad (2)$$

The resultant gravity force per unit volume is equal to

$$F_g = C_b g \left( \rho_s - \rho_f \right). \qquad (3)$$

The density of the solid particles typically exceeds the density of the fluid phase in transport of sediment particles on Earth's surface; so there $F_g$ is directed downwards to the bed (Figure 1a).

Turbulence is widely quoted as a support mechanism for suspended particles. We postulate here that pressure and viscous forces exerted by turbulence onto suspended particles near the bed must average over time to supply a force $F_{turb}$, directed away from the bed and equal in magnitude to $F_g$, for sediment to be suspended with an equilibrium near-bed particle concentration $C_b$. The magnitude of $F_{turb}$ will here be estimated from the scales of turbulent boundary layers. The basic idea that will be pursued is that observed accelerations of fluid parcels in turbulent flow are the expression of turbulent forces. Accelerations in wall bound turbulence are at the front of current developments in unsteady turbulent fluid dynamics (Yeo et al., 2010). Physical experiments (La Porta et al., 2001) and Direct Numerical Simulations (DNS) (Vedula and Yeung, 1999) investigating acceleration in homogenous isotropic turbulence have confirmed theoretical scaling of acceleration distributions proposed in the first half of the 20[th] century. However, near-boundary turbulence necessitates a more arduous analysis because the common neglect of the viscous term in the unsteady Navier-Stokes equation (La Porta et al., 2001; Vedula and Yeung, 1999) does not hold close to the boundary (Yeo et al., 2010). Awaiting these more rigorous developments, we follow the idea proposed by Irmay [1960; *Bagnold*, 1966], to evaluate the average acceleration experienced by a fictitious average fluid parcel as representative of the multitude of underlying turbulent movements. Surprisingly, this average acceleration has the same sign for both downwards (negative) and upwards (positive) turbulent velocity excursions, resulting in a net upwards time-averaged acceleration which is not equal 0 m/s$^2$, despite the time average of turbulent velocity fluctuations being, by definition, equal to 0 m/s. In essence, this average upward acceleration is a result of the impermeability condition (Day, 1990; Pope, 2000; Stokes, 1851). This condition necessitates that any upward directed turbulent motion must have been associated with an upward acceleration through time on a trajectory away from the boundary, where the vertical motion must have been zero. Reversely, any downward directed turbulent motion of a fluid parcel moving towards the boundary must experience a similar upward acceleration to cancel the downward motion upon arrival on the boundary (Figure 1b). This simple approach enables the estimation of a force scale from the Newtonian inference that the upwards acceleration is the expression of a net upward turbulent force ($F_{turb}$) acting per unit volume of fluid. The magnitude of $F_{turb}$ is now shown to follow from the scale-independent turbulence structure near a frictional boundary.

The velocity components (*u, v, w*) are directed along coordinate directions (*x,y,z*), and are here assigned to the stream-wise (*u*) and lateral (*v*) boundary-parallel velocities and the boundary-perpendicular velocity (*w*) respectively. The instantaneous, average, and turbulent velocity components are related as $(u,v,w)=(\bar{u},\bar{v},\bar{w})+(u',v',w')$, where the overbar denotes a time average, and the prime denotes the instantaneous turbulent velocity fluctuation. The time average of turbulent velocity components is, by definition, 0, and measures of the average intensity of turbulence are conventionally reported either as the mean of squared turbulence $\overline{(u',v',w')^2}$, or as the root-mean-square of turbulence $\sqrt{\overline{(u',v',w')^2}}$ or simply $(u',v',w')_{RMS}$. Velocity near the frictional boundary is appropriately normalized with the friction velocity scale $u^*$: $(u^+,v^+,w^+)=(u,v,w)/u^*$ and distance along coordinates near the boundary are normalized with friction velocity and kinematic viscosity $v$: $z^+=zu^*/v$. In this notation, the superscript $^+$ denotes non-dimensionalized velocity and length scales. The stream-wise velocity in turbulent shear flows collapses onto the "logarithmic law of the wall" for widely varying flow conditions under this normalization. This does not necessarily mean that the turbulence characteristics also collapse if normalized with the friction velocity. DeGraaff and Eaton (2000) note that such universal turbulence scaling is not supported by the body of available measurements. Townsend (1976) demonstrates, however, that similarity of turbulent motions is uniquely possible for the boundary-perpendicular component *w*, and not for the other components. This boundary perpendicular component has, indeed, been confirmed to collapse when normalized with the friction velocity (Figure 2; De Graaff and Eaton, 2000). The analysis presented here makes use of this collapse of $\overline{w'^2}^+(z^+)$, and its structure is therefore reviewed in detail.

We consider the scales of turbulence in particle-free flow over a smooth, impermeable boundary. The no-slip and impermeability conditions (Day, 1990; Pope, 2000; Stokes, 1851) require that fluid in contact with the boundary has no tangential or perpendicular velocity relative to the boundary. The boundary-perpendicular velocity component $w^+$ is therefore equal to 0 at $z^+=0$, and so is the turbulent component $\overline{w'^2}^+$. Immediately above a perfectly smooth boundary, at elevations of $z^+\ll1$, molecular diffusion dominates over convection and velocity fluctuations exhibit Brownian motion (Dreeben and Pope, 1998). In the viscous sublayer, further from the wall but below $z^+\sim5$, the stream-wise velocity increases as $u^+=z^+$. This is a region of two-component flow in x-y planes where boundary-parallel velocity fluctuations $(u'^+,v'^+)$ start to be established but the vertical fluctuations remain small (Pope, 2000). Fluctuations in the vertical velocity component increase rapidly only above the viscous sublayer ($\sim5<z^+$). A peak value in the turbulence intensity is reached at a distance of $z^+_{\mathrm{Imax}}\sim90\pm10$ from the boundary, and it remains quasi constant throughout the near-boundary flow (De Graaff and Eaton, 2000). In this analysis, we focus on the region of turbulent flow between $\sim5<z^+<\sim90$ that exhibits strong vertical spatial gradients in average boundary-perpendicular velocity fluctuations (Figure 2).

Slightly different values for the maximum intensity of vertical turbulence are reported in literature. The peak value attained at the turbulence intensity maximum is $\overline{w'^2}(z_{\mathrm{Imax}})\sim1.2u^{*2}$ or; $\overline{w'^2}^+(z^+_{\mathrm{Imax}})\sim1.2$. Spalart [1988], Nezu & Nakagawa [1993], and Townsend [1976] report $w'^+_{RMS}(z^+_{\mathrm{Imax}})\sim1.1$. Grass [1971] reports $w'^+_{RMS}(z^+_{\mathrm{Imax}})\sim1.0$, and DeGraaff and Eaton [2000] report $\overline{w'^2}^+(z^+_{\mathrm{Imax}})\sim1.35$. In this paper we use the numerical value of $\overline{w'^2}^+(z^+_{\mathrm{Imax}})\sim1.2\pm0.1$.

Equations of motion for spatial variation in velocity under spatially varying acceleration have the form:

$$\tfrac{1}{2}v(x)^2=\int a(x)dx \qquad (4)$$

Which can be written as

$$\tfrac{1}{2}\overline{w'^2}^+(z^+)=\int \bar{a}^+(z^+)dz^+ \qquad (5)$$

if applied to the average amplitude of the vertical turbulent motion.

Acceleration is here non-dimensionalized with viscosity and friction velocity as

$$a^+ = a \frac{\nu}{u^{*3}}, \tag{6}$$

which follows naturally from the conventional non-dimensionalizations of $z$ and $(u,v,w)$ introduced above.

No established functional form for the spatial distribution of acceleration near the boundary is readily available. This distribution is therefore established here, in a wholly empiric procedure, by fitting a functional form of $\overline{w'^2}^+$ to benchmark measurements and DNS results (Fig. 2), we find a good fit with a quadric relation:

$$\overline{a}^+(z^+) = C_1(90 - z^+)^2 \tag{7}$$

Equation (7) is substituted in Eq. (5) and the integration is performed

$$\overline{w'^2}^+(z^+) = -\tfrac{2}{3} C_1(90 - z^+)^3 + C_2 \tag{8}$$

The constants $C_1$ and $C_2$ are evaluated as $\frac{3}{2}\frac{1.2}{85^3}$ and 1.2 respectively from the boundary conditions of $\overline{w'^2}^+$ at $z^+=5$ and $z^+=90$, resulting in

$$\overline{w'^2}^+(z^+) = 1.2 - \frac{1.2}{85^3}(90 - z^+)^3 \tag{9}$$

Equation (9) has been plotted as the red line in Figure 2. The agreement between Eq. (9), DNS data, and measurements is

sufficient for the purpose of this analysis. The numerical values in Eq. (9) are the result of the fit to the boundary conditions at $z^+=5$ and $z^+=90$, the agreement of the shape of Eq. (9) and the observed universal distribution of $\overline{w'^2}^+$ is a pragmatic justification of the functional form assumed *a priori* in Eq. (7).

The present aim is to compare turbulent forces between $5<z^+<90$ to gravity acting on the suspended sediment. The average non-dimensional acceleration between $5<z^+<90$ is evaluated from Eq. (7) as

$$\left\langle \overline{a}^+ \right\rangle_5^{90} = \frac{1.2}{170} \tag{10}$$

Which, in combination with Eq. (6) yields

$$\left\langle \overline{a} \right\rangle = \frac{1.2 u^{*3}}{170 \nu} \tag{11}$$

$F_{turb}$ is now obtained by the Newtonian inference that the upwards acceleration is the expression of a net upward turbulent force acting per unit volume of fluid:

$$F_{turb} = \rho_f \left\langle \overline{a} \right\rangle = \frac{1.2 \rho_f u_*^3}{170 \nu}. \tag{12}$$

We now introduce the non-dimensional near-boundary suspension capacity parameter $\Gamma$ to compare the vertical turbulent forces $F_{turb}$ to the gravity force acting on suspended particles per unit volume $F_g$:

$$\Gamma = \frac{F_{turb}}{F_g} = \frac{\rho_f \; 1.2 u_*^3}{170 \nu g (\rho_s - \rho_f) C_b} = \frac{u_*^3}{140 \nu g \, R C_b}, \tag{13}$$

with $R = \rho_f / (\rho_s - \rho_f)$ being the relative density of sediment submerged in water. The numerical constant 140 derives from the scales of vertical turbulence discussed above [1.2/(2*(90-5))]. The propagated uncertainty of the estimation of turbulent scales from measurement is 140±20 (±15%). Note that the quantification of turbulent forces available to balance gravity forces in $\Gamma$ has been derived from clear-water scales. This approach will be justified in the following section.

**2.2 Interpretation**

The absence of particle size $d$ from Eq. (13) is a strong breach of the established intuition that grainsize is a primary control on particle suspension. The proposal of Eq. (13) is therefore a strong argument for a capacity perspective of particle suspension, as opposed to a competence perspective (Dorrell et al., 2013; Hiscott, 1994). Of course the dichotomy cannot be complete, and the role of grainsize in limiting the suspended particle concentration will be discussed in Section 4, following the

interpretation and discussion of the primary structure of the suspension capacity parameter.

When $\Gamma > 1$ the average vertical turbulent force in the flow exceeds the net gravitational pull on the suspended particles, the suspension is under-saturated. Such conditions might arise from a lack of availability of particles to suspended, either due to an absence of particles on the wall, due to particle size inhibiting entrainment from the wall, or due to cohesive forces keeping particles attached to the wall. The turbulent force $F_{turb}$ has been estimated from clear water turbulence kinematics above. This

clear water turbulent force is here interpreted as a force budget, derived from the shear of the overriding flow, that is available to either accelerate fluid or support particles in suspension. At under-saturated conditions, not all of the budget is needed to support suspended particles, and the remaining budget is utilised to generate turbulent accelerations. This interpretation therefore leads to the prediction that vertical turbulent kinematics are suppressed in the presence of suspended particles at concentrations below the saturation concentration. This interpretation is justified by both numerical as well as experimental

results. Direct numerical simulations of turbulent flows with suspended particles by Cantero and co-workers demonstrate that turbulence intensity near the boundary of the flow is decreased in the presence of particles for all components ($u,v,w$) at under-saturation (Cantero et al., 2009). This decrease in the statistical intensity of turbulence is linked to the decreased occurrence of hairpin vortices in those simulations (Cantero et al., 2009). Experimental evidence for the clear-water budget perspective presented herein is found in the experiments of Bennett et al. (2014). They measured turbulence in a mixing box with an

oscillating grid in the presence of various amounts of suspended particles. The measurements convincingly demonstrate that the presence of suspended particles reduces overall turbulent kinetic energy with respect to clear-water reference values without changing any of the boundary conditions. These observations in experiments and modelling justify the interpretation of the clear-water value of $F_{turb}$ as the system's total turbulent force budget. An apparent reduction in turbulence with increasing particle concentration is the result of a portion of this budget having been expended in maintaining particle suspension (Tilston,

2016). Both Bennett et al. (2014) and Cantero et al. (2012) invoke a similar expenditure of turbulent kinetic energy (TKE), arguing that part of the TKE is consumed or transferred from turbulent fluid to particles in order to keep them suspended. The interpretation presented here differs slightly in that we argue that the presence of suspended particles prevents part of the turbulence from ever being produced.

At $\Gamma = 1$, average turbulent forces in the near wall region of the flow are in equilibrium with the net gravitational pull on the

suspended particle load, this force balance prevents average net vertical acceleration of the sediment particles and the fluid between them. The flow is precisely saturated with suspended sediment near the boundary and $C_b$ can be seen as a saturation concentration. Equation (13) can thus be used as an analytical expression for near wall equilibrium concentration when $\Gamma$ is set to 1. It is suggested here that turbulence becomes precisely fully suppressed at saturation. This suggestion is supported by the complete suppression of turbulence production at high particle concentration in the mixing-box experiments of Bennett et

al (2014). Graf and Cellino (2002) report turbulence intensities measured in the presence of suspended sediment at saturation in a free surface shear flow. Turbulence intensities of both vertical and stream-wise components are reported to be significantly

suppressed close to the boundary of their experiments, but their experimental techniques did not allow confident assessment of turbulence at $z^+<90$.

When $\Gamma<1$, gravitational pull on the sediment dominates, and the flow does not have sufficient capacity to suspend all the particles present in the near-boundary region. The flow is over-saturated with sediment. This situation can be regarded from a particle perspective and a continuum perspective, both resulting in deposition of sediment from the base of the flow: firstly, from a particle perspective, particles will, on average, experience a wall-bound gravitational body force that exceeds turbulent pressure and viscous forces acting on the particle surfaces, and they will accelerate towards and settle onto the boundary; secondly, from the continuum perspective of the turbulent flow, the upward turbulent pressure and viscous forces are smaller than downward gravitational forces applied to the fluid by the particles, this prevents turbulent accelerations and results in turbulence extinction. Turbulence extinction at over-saturation must be expected to result in sedimentation as there is no mechanism countering gravitational settling of sediment. This second perspective is reminiscent of a recent breakthrough in DNS simulations of suspension flows (Cantero et al., 2009, 2011, 2012) that demonstrates how turbulence at the base of suspension flows is rapidly extinguished in over-saturated suspensions. The studies by Cantero and co-workers demonstrate that complete suppression of turbulence at oversaturation is related to the disappearance of the streak-vortices that form the legs of hairpin-vortices. These hairpin vortices are the dominant vertical structures in near-boundary turbulence (Adrian, 2007; Smith and Walker, 1995; Zhou et al., 1999), and their legs normally occupy the zone between $5<z^+<90$. Their suppression signifies a complete shutdown of the production of near-boundary turbulence, leading to rapid laminarization extending far beyond the normal viscous sublayer thickness. It is important to note that this laminarization is controlled by the ratio between concentration, flow conditions, and material properties as expressed in Eq. 13. This means that particles of quartz sediment in water will cause full turbulence extinction at low absolute concentrations for gentle flow conditions: the turbulent carrying capacity can be overloaded at low absolute concentrations. Cantero et al. (2011) stressed this occurrence of turbulence extinction at low absolute concentration and pointed out that this is contrary to the conventional assumption that sediment oversaturation is a high-concentration phenomenon.

### 3 Comparison of $\Gamma$ to measurements and previous formulations

#### 3.1 Measurements

$\Gamma=1$ for quartz particles in water at 10° C is plotted in Figure 3a, together with measurements of suspended particle concentrations. The graph supports the notion that any suspension flow with a suspension capacity parameter smaller than 1, which corresponds to the region above the $\Gamma=1$ iso-line in Figure 3, results in rapid deposition from the base of the flow, and a return to capacity transport. Data from studies with saturated suspensions (Bennett et al., 1998; Cartigny et al., 2013; Coleman, 1986; Graf and Cellino, 2002; Smith and McLean, 1977) lie around $\Gamma=1$.

$\Gamma=1$ also envelopes the upper range of data points from Vanoni (1940), Einstein and Chien (1955; reported in Montes Videla, 1973), Ordoñez (1970; reported in Montes Videla, 1973), and Montes Videla (1973). These measurements represent under-saturated suspensions in experiments where the flume floor consisted of smooth-glass or was covered by glued down sand-particles. Both conditions avoided formation of loose granular beds. Therefore, "it is doubtful if enough material was ever available to completely load the flow" (Vanoni, 1940).

#### 3.2 Previous formulations

The turbulence extinction threshold of Cantero et al. (2009, 2011, 2012) is virtually identical to $\Gamma=1$ (Figure 3b). The simple form of Eq. (13) has major advantages, however, over the threshold proposed by Cantero et al., which necessitates parameterizations using flow depth, sediment grainsize and settling velocity, and has bulk-flow-Reynolds-scale dependency (see Eqs. A.1.1-4). Appendix A.1 contains details on choices made for the parameterization of Cantero et al.'s threshold

condition in Figure 3b. The close correspondence with $\Gamma$=1 indicates that the here proposed force balance of the suspension capacity captures the mechanism of the underlying full suppression of hairpin-vortex turbulence generation observed in DNS experiments (Cantero et al., 2009, 2011, 2012).

It has not been previously resolved whether the physically appropriate boundary condition of a suspension field should be a summation of the inward and outward fluxes through the boundary, or a boundary concentration. Many multiphase modelling approaches define a sedimentation flux towards the flow boundary dependent on sediment concentration and the stagnant-water terminal settling-velocity $w_s$; and an entrainment flux away from the flow boundary that is empirically related to flow conditions. Such flux-based equilibrium can be compared directly to the near-bed saturation concentration in the phase space of Figure 3; which is illustrated by equating the sedimentation flux to an often-used empiric entrainment flux (Garcia and Parker, 1991, 1993)

$$C_b w_s = \frac{AZ^5}{1+\dfrac{AZ^5}{0.3}} w_s, \tag{14}$$

where $A$ is an empiric constant and $Z$ is a non-dimensional parameter that empirically depends on $u^*$, particle size and density, fluid density and viscosity, and gravity. The empiric flux-based approach has been parameterised for quartz particles with a diameter of 150 μm grains in water, which yields results that are similar to the present saturation concentration theory (Figure 3b).

Empiric relations have been used as closures for suspension concentration fields in sediment transport budget calculations in the absence of theory. Figure 3c compares the present theory with a number of proposed formulations, for the case of 150 μm sand (see Appendices A.2-4 for details of these formulations and the parameterizations used). The empiric relations have a similar general signature as the theory, but the capacity theory outperforms the previous relations when the predictions of measurements is considered (Figure 3c).

## 4 Particle size

Particle size $d$ is absent from the suspension capacity parameter, while it is often regarded as a main control on particle suspension. Essentially, this expectation arises because larger particles settle faster under action of gravity. This leads to an intuitive incorporation of the settling velocity $w_s$ in assessment of particle suspension transport, mostly in a non-dimensional ratio with the friction velocity following Bagnold (1966), or in the non-dimensional Rouse number following Rouse (1937). This incorporation is justified in the region far above the bed, where turbulent diffusion and vertical settling are the competing processes that determine the particle concentration. This intuitive approach is not satisfactory, however, in the near-boundary zone where terminal settling velocities in stagnant water cannot be justified to be the main controlling parameter as turbulent structures of the size of particles are associated with turbulent accelerations that may exceed $g$ (Bagnold, 1966; Irmay, 1960; La Porta et al., 2001). Also, note how $w_s$ is immediately dropped from Eq. (14) to yield a balance between erosion and deposition that relates the basal concentration to the friction velocity, quantitatively approaching $\Gamma$=1 (Figure 3b). Grain size independence of particle transport near the boundary is also encountered in the initiation of motion of fine grains under hydraulic smooth conditions. In those conditions the critical Shields parameter plots as a straight line with negative slope in the Shields diagram. This indicates that initiation of transport is controlled by the constant product of the critical Shields parameter ($\theta_{cr}$) and particle Reynolds (Re*) number. The result is a parameter for the initiation of motion in hydraulically smooth conditions that lacks the presence of $d$, and that resembles $\Gamma$:

$$\Theta_{cr} Re^* = \frac{u^{*3}}{Rg\nu} = K \tag{15}$$

A query of the Shields diagram reveals that $K \sim 0.11$. The threshold of particle motion is invariably perceived as a competence concept; it analyses whether the bed shear stresses are strong enough (i.e. competent) to initiate movement of grains of specified size. However, the preceding analysis shows that under hydraulically smooth conditions, all grainsizes are mobilised at the same absolute level of dimensional shear stress.

The Stokes number, rather than the Rouse number, is an appropriate measure for the grainsize dependent differential motion between particles and turbulent structures near the boundary. The Stokes number is the non-dimensional ratio of particle relaxation time and a characteristic hydrodynamic timescale. The particle relaxation time is

$$T_{particle} = \frac{\rho_s d^2}{18 \rho_f \nu} .$$

(16)

The characteristic timescale of the hydrodynamic setting at hand is determined from the equations of motion of the fictitious

average parcel in the near boundary region. The timescale for the acceleration of the fictituous average parcel is used as the hydrodynamic timescale:

$$T_{hydrodynamic} = \frac{\Delta \hat{w}_{RMS}^{+}}{\langle \bar{a} \rangle} = \frac{170 \nu}{\sqrt{1.2 u^{*2}}} ,$$

(17)

and the Stokes number for the problem at hand is

$$St = \frac{\rho_s d^2}{18 \rho_f \nu} \bigg/ \frac{170 \nu}{\sqrt{1.2 u^{*2}}} .$$

(18)

Particles with $St \ll 1$ are responsive to viscous forces exerted by the surrounding fluid, tend to follow turbulent movements of the fluid parcels in the near boundary region, and will attain saturation concentrations predicted by $\Gamma = 1$ irrespective of their grain size. Trajectories of larger particles with $St > 1$ will not mimic the fluid flow path and behave ballistically. Such ballistic behavior will cause a lag both in acceleration away from the bed in upward turbulent excursions and deceleration in downward turbulent excursions. The former will result in entrainment limited suspension, the latter in enhanced deposition, and both

cause concentrations of larger particles to be below the predictions of $\Gamma = 1$. Figure 4 displays the Stokes number for quartz particles of different sizes in the silt to fine-grained-sand classes. Particles of $d < 250$ μm have $St < 1$ in water for the range of friction velocities ($u^* < 0.15$ m/s) typically encountered in natural flows. The implication is that particle size has no first order effect on suspension capacity for quartz particles smaller than 250 μm in water. Supporting measurements can again be found in the work of Bennet et al. (2014). Turbulence is fully extinguished in their experiments in the presence of high concentrations

of 150-200 μm quartz particles. Contrastingly, turbulence is suppressed merely 10% in the presence of 1.0-1.2 mm particles. Steady state concentrations of those larger particles are, indeed much lower than the concentrations attained for 150-200 μm particle sizes, suggesting that the turbulent particle support mechanism rapidly becomes inefficient at $d > 200$ μm.

## 5 Boundary roughness and elevated particle concentration

Simplifications have been made in the establishment of our suspension capacity parameter. These simplifications were necessary for the initial establishment of the work. While it is possible that careful consideration of neglected effects leads to future modifications of the suspension capacity parameter; the first order structure of $\Gamma$ already captures the first order structure of available data (Figure 3). Incorporation of second order considerations should not come at cost of increasing the intricacy of parameterisation, as the simplicity of $\Gamma$ forms a key rationale for its development and ensures a broad application in practical

problem solutions. One of the strengths of $\Gamma$ in Eq. (13) is that it can easily be calculated without need to constrain boundary conditions that are difficult to estimate in field conditions or impossible to constrain in the absence of measurements. Nevertheless, we comment on the implications of two of simplifications here.

## 5.1 Roughness

The turbulent force scale $F_{turb}$ was derived from reported distributions of turbulence near smooth boundaries. A bed of sediment particles will form the boundary to flows in nature. This particle bed will have a surface roughness that may impact the structure of turbulence if the roughness scale is sufficiently large and particles extrude through the viscous sublayer. The rugosity of silt and very fine grained sand will generally present hydraulically smooth boundaries to flow, while fine grained sand may be transitionally rough for most of the flow conditions considered in this paper (García, 2008). Specifically, the roughness scale $k^+$ is smooth for all values of $u^*$ considered herein for particles 25 μm and smaller. For fine grained sand with a 200 μm particles size, $k^+$ is 2 at a low $u^*$ (~0.01 m/s), becomes transitionally rough ($k^+>3$) at $u^*=0.015$, and is equal to 24 at the highest values for $u^*$ considered herein (0.12 m/s). It is important to evaluate how do these roughness scales impact the analysis presented above. The main consideration should be whether the scales of $\overline{w'^2}^+$ attained at specified values of $z^+$ are affected by roughness. The peak value of $\overline{w'^2}^+ =1.2$ $u^*$ is not affected by boundary roughness in the range of $k^+$ 0-85 reported by Grass (1971; his Fig. 5) or 0-136 as reported by Nezu and Nagakawa (1993; their Fig. 4.8). Jimenez ( 2004) reviews turbulence near rough boundaries and concludes that the plateau value for $\overline{w'^2}^+$ above rough boundaries scales similarly to the smooth cases reviewed by Graaf and Eaton (2000). Also, the peak value is attained at a similar level above the boundary in the measurements of Grass (1971). The data collected by Grass do show that the gradients of $\overline{w'^2}^+$ are not equal between smooth, transitionally rough, and rough cases throughout the near-boundary region. With increasing roughness, $\overline{w'^2}^+$ values are elevated closer to the boundary. This signifies that turbulence levels are higher relatively low in the flow, where turbulent eddies are shedding off particle irregularities. To compensate, gradients are lower higher up in the region $z^+<90$, such that values attained at the peak elevation are similar. This will result in similar layer-averaged values for $F_{turb}$ and hence for $\Gamma$ and $C_b$ compared to smooth cases.

## 5.2 Support mechanisms other than fluid turbulence at high particle concentration

Turbulence is generally accepted to form the sole particle support mechanism in the limit of low particle concentrations. As particle concentration increases, however, pathways of particles will increasingly occupy trajectories that carry the particles into each other's vicinity, and particle-particle or particle-fluid-particle interactions may become prevalent in what is termed the "sheet-flow regime" of sediment transport. Bagnold (1954) first considered that the forces exerted on particles under shear by particle-particle collisions could be integrated over a surface area to obtain a "dispersive pressure" that may support particles in the sheet-flow regime. It is beyond the scope of this paper to review the body of research of particle support in high-concentration shear that has since been developed, yet two successful approaches will be mentioned here. Firstly, a theory in analogy to the kinetic theory of gasses was developed by Jenkins and Hanes (1998) for particles that are involved in intermittent elastic collisions. Secondly, for even higher particle concentrations, a frictional rheology has been developed for particles that are in in quasi-enduring contact (Jop et al., 2006). While both these high-concentration analytical approaches have initially been developed for particle-interactions in the absence of fluids, they have been modified and extended to account for the presence of viscous and turbulent pore fluids to satisfactory extent (Cassar et al., 2005; Hsu et al., 2004). These high-concentration regimes do not play a role in sediment transport at low to moderate friction velocities. Specifically, sheet-flow thickness falls below the minimum thickness of 3 particle diameters at a Shields parameter value of ~0.7 (measurements of Sumer et al., 1996, as processed by Hsu et al., 2004), which in the case of the 150-200 μm particles considered dominantly throughout this paper translates to a friction velocity of ~0.05 m/s. Again, it must be stressed that turbulence extinction due to over-saturation with suspended particles is not reserved for high absolute concentrations, but occurs at low concentration for gentle flow conditions (Cantero et al., 2011). Therefore, the suspension capacity parameter $\underline{\Gamma}$ applies in a considerable range of flow conditions where sheet flow is absent. However, at the most energetic scale of suspension $\Gamma$ predicts saturation

concentrations going up to levels that are firmly in the collisional and frictional regimes, where it is established that particle-interaction dynamics, rather than fluid turbulence supports particles. In these high basal sediment concentrations at intense shear, we envisage stacked regimes of particle support. In a fully developed basal region these could include from the bottom upwards: a static bed of particles; a frictional regime of sliding grains; a collisional regime; a dynamic turbulent support regime, where vertical fluid motions become established; and a turbulent dispersion regime. Our manuscript focusses on the dynamic turbulent support regime.

## 6 Summary discussion and conclusions

We have introduced a non-dimensional suspension capacity parameter $\Gamma$, which compares the gravity forces acting on the near-bed suspended particles to the vertical turbulent force acting near the base of a turbulent flow. $\Gamma$ is general for particles in a turbulent viscous fluid near a boundary, and while focus here is on applications to water-born transport of natural sediment particles near the base of rivers and above the seafloor, it can be applied to a wide variety of multiphase transport problems in terrestrial and extra-terrestrial flows. It needs only be applied to regions within flows where turbulent stress gradients are large. In many problems, this region extends only a few millimetres from the flow boundary, and the present theory is therefore especially useful as concentration boundary condition at flow boundaries in simulations (e.g. Kane et al., 2016). Dynamic suspension support by turbulent stress gradients can generally be neglected in the bulk of the turbulent fluid, because turbulent stress gradients are much smaller above the turbulence intensity maximum (Figure 2); there, turbulent dispersion modelling using drift-velocity or drag approaches (Basani et al., 2014; McTigue, 1981), or the Rouse number suffice. Figures 3b-c yield justification for model simulations utilizing Eq. (14) or other empiric closures (Garcia and Parker, 1991, 1993; van Rijn, 1984; Smith and McLean, 1977; Zyserman and Fredsoe, 1994) for the near-wall boundary condition, but it is suggested here that a saturation concentration $C_b$ calculated with the suspension capacity parameter $\Gamma=1$ can be used as an appropriate boundary condition in future work. The suspension capacity parameter Eq. (13) does more justice to available concentration measurements, and eliminates the need to set values of non-physical empiric parameters. When $\Gamma$ is not equal to 1, the suspension is under- or over-saturated with particles. Adjustments to the disequilibrium in such situations may prove to yield more significant differences between the near-bed-capacity-concentration and entrainment-vs-settling-flux approaches. A key aspect to consider is which variables will govern the timescale of saturation, and which govern a timescale of relative adjustment through erosion and settling fluxes. The turbulent kinematic scales involved in adjusting to saturation increase as $u*^2$, and the thickness of the saturation region goes as $1/u*$. We therefore predict that saturation timescales in the dynamic suspension support regime decrease rapidly with increasing vigorousness of sediment transport, as the region under consideration becomes smaller and the kinematic rates increase rapidly. The opposite is true for non-steady adjustments through settling. Settling timescales are independent of the vigorousness of shear and turbulence for relatively large particles or modest turbulence. In such conditions, the adjustment time scale through settling depends on the settling velocity and the vertical section that needs to re-equilibrated (Dorrell and Hogg, 2012; Ganti et al., 2014). At vigorous turbulent shear conditions, however, the mixing action of the turbulence induces a vertical dispersion flux that cannot be ignored. The turbulent mixing flux is positive in the direction of decreasing sediment concentrations, and therefore directed opposite to the settling flux caused by gravity. This mixing flux therefore causes a delayed adjustment with respect to the pure settling case because it acts against the settling flux. The timescales of adjustment to disequilibrium in a settling flux approach consequently increase with increasing vigorousness of turbulent shear (Dorrell and Hogg, 2012). Saturation and adjustment length and timescales to unsteady conditions can thus provide experimental tests for the validity of the near-bed suspension capacity concentration perspective over the entrainment-settling fluxes perspective. It is emphasised here that the saturation length and timescales for dynamic support are relevant to the region $z^+<90$ only. Adjustment scales of particle transport in bedload have recently been shown to be fundamental to creating ripple and dune bedforms in particulate beds. Determination of near-bed suspension

saturation scales could elucidate the fundamental formational mechanisms of bedforms that are impacted by highly unsteady changes in suspension transport, such as anti-dunes and cyclic-steps (e.g. Cartigny Sedimentology experiments). The length and timescales derived by Dorrell and Hogg (2012) for the combination of settling due to gravity and mixing due to turbulence govern adjustments in the bulk of the flow away from the bed.

Particle size is absent from the suspension capacity parameter, but it does appear in the Stokes number. Large particles with $St \gg 1$ will start to behave ballistically and travel straight through turbulent eddies without following the turbulent accelerations of the fluid. The concentration of such ballistic particles must be expected to be lower than the $C_b$ predicted for $\Gamma=1$. $St$ is lower than 1 for clay, silt, and very-fine- and fine-grained sediment particles in water, under all reasonable turbulent conditions (Figure 4). This means that for the vast majority of sediment, grainsize bears no influence on how much sediment can be

transported in suspension by water close to flow-bed interfaces on the Earth's surface. Gravitational acceleration $g$ is absent from $St$, as particle size $d$ is absent from $\Gamma$. Therefore, gravity and particle size are not combined in either of the two non-dimensional groupings that determine the suspension capacity and particle-size dependence of suspension near the flow-bed interface. This must mean that the kinematic scale of settling velocity under gravity $w_s$ is an irrelevant variable for the problem of turbulent suspension capacity in the near-wall region. Both the independency from particle size and irrelevance of the

settling velocity are only valid for the thin layer below $z^+=90$, where vertical turbulence gradients dominate particle suspension. Turbulent dispersion dominates higher up in the flow, and grainsize and vertical settling velocity are important controls there.

**Appendix A Previously proposed empirical relations for the near boundary particle concentration**

**A.1 Cantero et al. (2009, 2011, 2012)**

Cantero et al. (2009, 2011, 2012) have empirically determined a threshold above which turbulence in flows at the base of

their DNS experiments is fully suppressed

$$\frac{\text{Ri}_\tau \widehat{w_s}}{\text{K}_c\{Re_\tau\}} = 1 \frac{Ri_\tau w_s}{\mathcal{K}_c\{Re_\tau\}} = 1 \ , \tag{A.1.1}$$

where the subscript $\tau$ denotes the shear-Richardson and shear-Reynolds flow scales, defined as

$$Ri_\tau = \frac{g(\rho_s/\rho_f - 1)CH}{u^{*2}} \ , \text{ and } Re_\tau = \frac{u^*H}{\nu} \ , \tag{A.1.2}$$

where $C$ is the average sediment concentration, and $H$ the flow size perpendicular to the wall. The settling velocity $w_s$ has been

made non-dimensional by

$$w_s = \frac{w_s}{u^*} \ . \tag{A.1.3}$$

And the weak dependence of the turbulence threshold value on the Reynolds number is suggested to be (Cantero et al., 2012)

$$\mathcal{K}_c = 0.041\ln(Re_\tau) + 0.11 \ . \tag{A.1.4}$$

The term $Ri_\tau w_s$ has a structure that is very similar to $1/\Gamma$:

$$Ri_\tau w_s = \frac{g(\rho_s - \rho_f)CHw_s}{\rho_f u^{*3}} \tag{A.1.5}$$

This means that a direct comparison between $\Gamma$ and the turbulence suppression in DNS is possible when the turbulence extinction threshold is parameterized with appropriate values for $H$ and $w_s$. The plotted line in Figure 3b has been parameterized with the scales of the experimental comparison proposed in Table 2 of Cantero et al. (2012). The ratio of near-

boundary concentration and depth-averaged concentration at turbulence suppression has been set to 3.45 in accordance to Figure 1 of Cantero et al. (2012).

## A.2 Smith and McLean (1977)

Smith and McLean (1977) suggest

$$C_b = \frac{C_{max}\gamma_0 T}{\left(1+\gamma_0 T\right)} \tag{A.2.1}$$

for the functional form for the sediment concentration at the reference level, where $C_{max}$ is the maximum sediment concentration, $T$ is the transport stage parameter (van Rijn, 1984, 1993), and $\gamma_0$ is an empirical constant $O(10^{-3})$. Choices for the appropriate values of these parameters have been made so as to follow the original publication (Smith and McLean, 1977). $C_{max}$ is set to 0.65 (Smith and McLean, 1977). The claimed appropriate value for $\gamma_0$ varies; the original publication reports fitted and computed values between 1.9 and 2.4, while a value of 4 is also reported (van Rijn, 1993). Here, the original concentration measurements for 270 μm sand are used to calculate the value of $\gamma_0$ that makes Eq. (A.2.1) satisfy each original measurement individually:

$\gamma_0 = [0.0033; 0.0087; 0.0034; 0.0037; 0.0030; 0.0030; 0.0030]$.

The average value is 4.0e-3, and this is used in Figure 3c.

## A.3 Zyserman and Fredsøe (1994)

Zyserman and Fredsøe (1994) suggested

$$C_b = \frac{A\left(\theta - \theta_{cr}\right)^n}{1+\dfrac{A\left(\theta - \theta_{cr}\right)^n}{c_m}} \tag{A.3.1}$$

as an empiric relation between suspended particle concentration and flow conditions, where $\theta$ is the Shields parameter. The critical Shields parameter $\theta_{cr}$ has been calculated with an explicit analytical formulation (Cao et al., 2006). The value of the empiric parameters as suggested in the original publication (Zyserman and Fredsoe, 1994) are used in Figure 3c: $A$=0.331; $n$=1.75; and $c_m$=0.46.

## A.4 van Rijn (1984)

van Rijn (1984) suggested for the near boundary particle concentration

$$C_b = \frac{0.035 d T^n}{\alpha_2 z_a d^{*m}}, \tag{A.4.1}$$

where $d$ is the median particle size, and * denotes a non-dimensionalisation with viscosity, density and gravity scales. The empiric constant and exponents are here used as suggested in the original publication (van Rijn, 1984): $\alpha_2$=2.3; $n$=1.75; and $m$=0.3. A number of different ways have been proposed to set the elevation of the reference level $z_a$, the elevation of $z^+$=90 is used here.

## 7 Acknowledgements

This is Kavli Institute of Theoretical Physics at UCSB report NSF-KITP-16-098, supported in part by the National Science Foundation under Grant No. PHY11-25915. JTE acknowledges the organisers and attendees of KITP GeoFlows2013 and the Max Planck Institute for Physics of Complex Systems GeoFlo2016 for discussions. This contribution is part of the Eurotank

Studies of Experimental Deepwater Sedimentology (EuroSEDS), supported by the Netherlands Organisation for Scientific Research (grant# NWO 864.13.006), ExxonMobil, Shell, and Statoil. Reviews by two anonymous referees and discussions with Mike Tilston helped us to clarify our ideas. One of the anonymous referees attended us to the similarity in structure between our suspension capacity parameter and the grain size independent threshold of motion for particles under hydraulically

smooth conditions (Eq. 15).

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

**Figures**

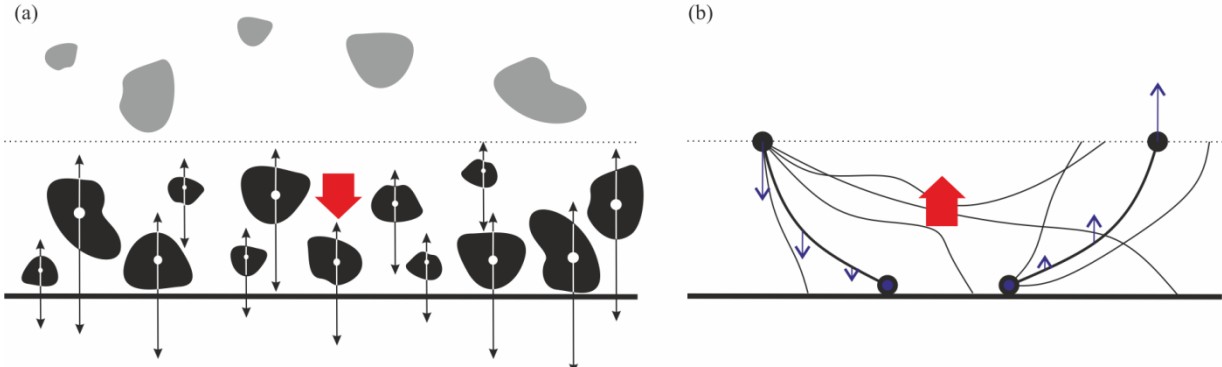

**Figure 1. Schematic diagram of gravity and turbulence acting on the suspension within the near-boundary region.(a) Each particle is acted upon by a downward-directed gravitational force and upward directed buoyancy force (black arrows). Particles in suspension above the near-boundary region, indicated in grey, are not considered herein. The red arrow indicates $F_g$ acting on the suspended particle load within the near-boundary region. (b) The vertical turbulent velocity $w'$ (blue arrows) is 0 m/s at the impermeable no-slip boundary at the base of the suspension, and increases throughout the near-boundary region. Parcels of suspension moving upwards or downwards through the top of the near-boundary region are subject to a multitude of possible velocity evolutions (thin black lines), which can be represented by the pathways of "fictitious average parcels" (thick black lines; (Irmay, 1960). The blue arrows indicate the vertical component of velocity; parcels with decreasing downwards velocities and increasing upwards velocities both experience upwards acceleration. The red arrow indicates $F_{turb}$ derived from this average upwards fluid acceleration.**

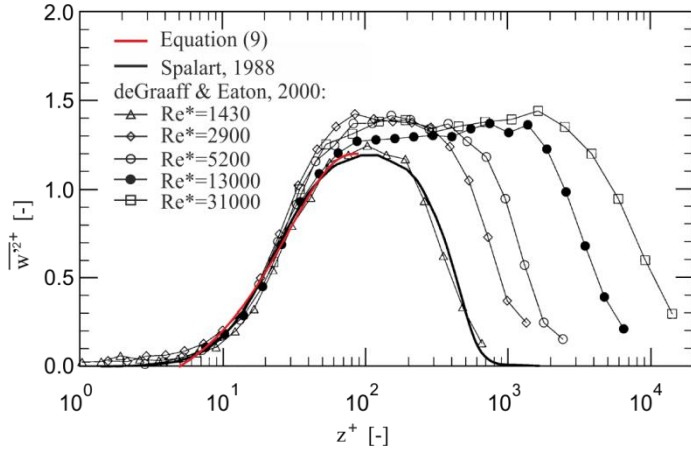

**Figure 2. Equation 9 compared to universal scaling of the vertical turbulent velocity in the near-boundary region in DNS (Spalart, 1988) and physical experiments (De Graaff and Eaton, 2000). Equation (9) is plotted as a red line. Modified after De Graaff and Eaton (2000).**

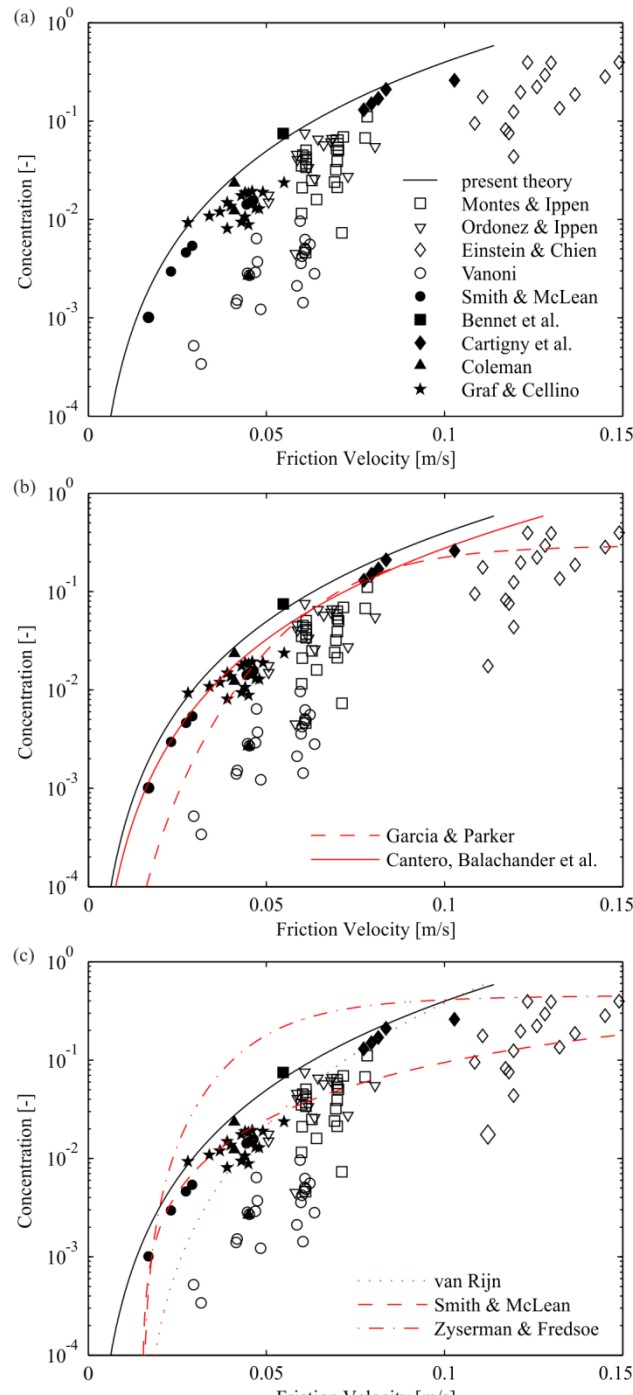

**Figure 3. The suspension parameter for saturated suspensions ($\Gamma$=1) plotted in the space defined by the friction velocity and the near-boundary particle concentration. (a) Comparison of theory to measurements of saturated suspensions and measurements of under-saturated suspensions. (b) Comparison of theory to a threshold for turbulence extinction observed in DNS experiments (Cantero et al., 2009, 2011, 2012), and balanced sedimentation-entrainment fluxes (Garcia and Parker, 1993). (c) Comparison of theory to suggested empiric relations.**

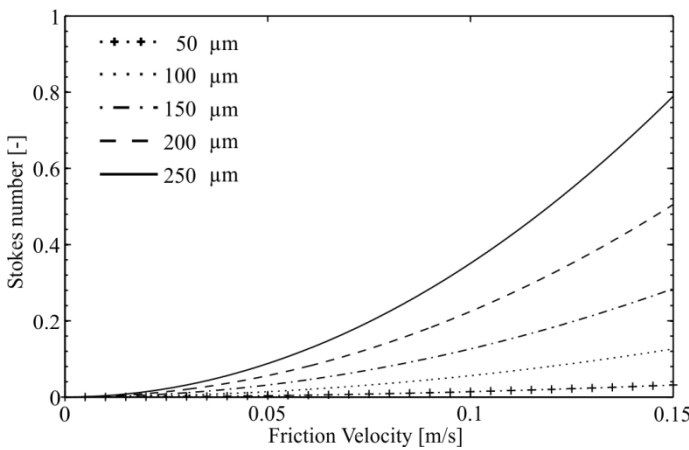

**Figure 4. The Stokes number (Eq. 18) of different sized quartz particles in the near-boundary region.**