# Peer review of "Physical theory for near-bed turbulent particle-suspension capacity."

_Earth Surface Dynamics, 2016_

## Referee Comment (RC1) · Anonymous Referee #1 · 11 Oct 2016

This is an interesting paper detailing a quasi-theoretical model for near bed suspended sediment capacity, based on a force balance. However, the authors must address the suitability of the model chosen, and the quasi-theoretical method by which it is derived, to the flow it is applied too.

The logic between balancing the upwards force and downwards gravitational force to determine near bed mass loading is sound. However, the assumption that the upwards force balance can be approximated by turbulent forces (derived from clear-water flow numeric and experiments) in this region is not. As the authors note this region is likely super saturated, dampening turbulent fluid motion. Moreover, in high concentration (here near bed) regions the fundamental mechanics for supporting sediment is not turbulent motion, rather particle-particle interactions (Jenkins & Hanes, 1998; Hsu, Jenkins & Liu, 2003; Berzi & Jenkins, 2008; Yu, Hsu, Jenkins & Liu, 2012). It is not clear

why the assumption that clear-water dynamics approximates the mechanics sediment suspension is valid.

Aside from this, there are a few minor issues to address: pg1) L18-20 There have been many empirical and theoretical models of sediment transport capacity.

pg1) L23-25 The absolute capacity is determined by a diffusion approach (depending on the continuum / discrete model approach used) but must be appropriately closed. The reference concentration is a non-unique means to achieve this.

Pg3) L25 Not necessarily true for particle laden flow and permeable beds.

Pg4) can you give typical range of z imax in an open channel flow in comparison to the sheet-load layer thickness.

Pg4) can you expand this argument for clarity.

Pg4) Eq 7 – why is a quadratic formula justifiable (other than an empiric fit to data). The numeric and empiric evidence presented suggests that an exponential function would be more appropriate. If clear water flow dynamics are appropriate can this be derived from dynamics in the log-law regime predominately covered by the model.

Pg5) L2-5 Clarify what is meant by a "strong" agreement – what is the error, how does this compare to alternative formulations?

Pg5) L19 suggest writing 140$\pm$20 (15%) for clarity.

Pg6) L9-15 How quickly does concentration change fundamental flow dynamics (near bed is likely of high concentration).

Pg6) L30-35 There are many alternative mechanisms for sediment suspension beyond turbulence saturation, e.g.: particle-particle interaction (granular diffusion) and matrix support, as evident in sheet layer and debris flow.

Pg7) L18 suggest delete red line

Pg8) L1 and L14 Note Z depends on d and therefore so does Cb (after eliminating ws)

Pg10) L10 Saying particle size bears no influence on capacity in flows near to the earths surface is far to vague. You must put this in context as at the flow-bed interface, and the particle size dependence of the remainder of the flow.

Pg12-14) References are incomplete and inconsistently formatted.

---

## Author Comment (AC1) · 14 Oct 2016

We acknowledge the referee's thoughtful comments.

The main comments of referee 1 touch on two important issues:

a) The manuscript does not address the relation between our criterion and treatments of high concentration basal sediment layers. Specifically, the referee points towards a series of established papers that extend the similitude between granular collisional flow and the kinetic theory of gasses to high-concentration collisional-sheetflow transport under strong shear (Jenkins and Hanes 1998 and following papers). Concentrations are high enough in this regime so that binary collisions between two particles take place constantly, yet not high enough to create enduring frictional contacts between grains, or force chains connecting many grains. Vertical gradients in particle collisions

within these high-density particle-fluid mixtures supply the upward forces to counter gravity forces in that analytical framework. This makes the referee question our choice of turbulence as the main support mechanism at the base of flow.

In answer to this comment it is important to stress the main achievement of Cantero et al.'s DNS work, which our manuscript strongly relates to. They established that the turbulent carrying capacity can be overloaded at low absolute concentrations. Turbulence extinction due to sediment oversaturation can consequently take place at low concentrations. This is contrary to the conventional assumption that sediment oversaturation is a high-concentration phenomenon.

A simple answer to the referee's comment is thus that our criterion is valid where the assumption that turbulence is the main support mechanism holds. In first instance, at low shear, this is true all the way down to the static bed. At high values of shear, and high basal sediment concentrations, we envisage stacked regimes of particle support. In a fully developed basal region these could include from the bottom upwards: a static bed of particles; a frictional regime of sliding grains; a collisional regime; a dynamic turbulent support regime; and a turbulent dispersion regime. Our manuscript focuses on the dynamic turbulent support regime

We recognize that the extension of our work to high-concentration basal regimes is an important issue to address in the future, but the integrated treatment of stacked static-frictional-collisional-turbulent support regimes is beyond the scope of this manuscript.

b) Secondly, there is a request for justification of the use of clear–water turbulent scales to determine the maximum concentration of sediment that can be suspended in what is, evidently, therefor not a clear-water situation.

The use of clear-water conditions as a scale for the vertical turbulent force is a pivotal issue in our manuscript. It is discussed on page 6, lines 7-23; but perhaps we should have made the importance more clear by giving this issue a more prominent role in the manuscript. Informally worded, I would say that the clear water turbulent scales

**ESurfD**
indicate the budget that is available for the combined dynamic support of sediment and turbulent accelerations. In clear water all of this budget is applied to turbulent accelerations. Part of the budget is used for particle support in undersaturated conditions, this leaves a limited budget left-over to accomplish turbulent accelerations. These should thus, in presence of supported sediment, be suppressed with respect to the clear water condition. The papers by Graf and Cellino (2002) and Cantero et al. (2009) are cited as support. The saturation condition is determined by equating the gravity force acting on the basal sediment concentration to the clear water turbulent force scale (i.e. Gamma=1).

We will strive to clarify the discussion of these important issues in the revised manuscript.

The minor issues can be clarified or corrected in the revision.

---

## Short Comment (SC1) · 17 Oct 2016

Regarding the use of turbulence characteristics from the near bed region of clear-water shear flows in the force balance to determine capacity versus competency driven depositional processes in particle laden density flows: I would like to direct the authors to a recent article by Bennett et al. (2013) entitled "Turbulence suppression by suspended sediment within a geophysical flow". In it, the authors employ a well-known experimental set-up to study the effects of suspended sediments on the flow's turbulence characteristics. In it they make the observation that the flow's turbulent kinetic energy (TKE) is inversely proportional to suspended sediment concentration but, importantly, its turbulent length scales ( $\lambda \equiv k^{(3/2)/\varepsilon}$ ) and time scales ( $\tau \equiv k/\varepsilon$ ), where k and  $\varepsilon$  are the turbulence production and dissipation terms, are constant for all suspended sediment concentrations. The authors conclude that the apparent loss in turbulence production

results from converting the fluids kinetic energy (or sediment suspension potential) to the kinetic energy of the suspended particles. Taking this thought further, it implies that clear-water values of TKE represent k (the systems total turbulence production) and the "apparent" reduction in TKE with increasing suspended particle concentrations is the result of a portion of k having already been expended in maintaining particle suspension. In theory, turbulence should be fully damped with higher particle concentrations, not because its production has been decreased, but because all of the turbulent energy produced by the system is used exclusively for the purposes of maintaining sediment suspension, and the system is carrying its maximum potential suspended load.

In the context of the present manuscript, the above view on the relationship between TKE and suspended sediment concentration appears to justify the use of clear-water turbulence statistics in their force balance. Here, the authors state that capacity driven deposition occurs when F turb-F g=0, which can be interpreted to read as the apparent turbulent forces (F (ãĂŰturbãĂŮ app)) can be described as F (ãĂŰturbãĂŮ app )=F turb-F g. In other words, the flow's measurable turbulence characteristics are the result of its natural (clear water) turbulence, which is then modulated by the particles suspended in the flow. In this way, their argument is entirely consistent with the observations of Bennett et al. (2013), and maximum sediment concentrations are achieved when F\_(ãĂŰturbãĂŮ\_app)=0 (i.e. flow is artificially laminarized by it's suspended sediment load). As such, I would commend the authors on deriving such an elegant (and deceptively simple) approach that can equate the onset of capacity driven deposition to the cessation of fluid turbulence, and I would highly recommend incorporating the above observations into the discussion to alieve any skepticism that might exist on the use of clear-water turbulence parameters in describing the suspension potential of particle laden flows.

Sincerely, Mike Tilston

**ESurfD**

---

## Referee Comment (RC2) · Anonymous Referee #2 · 4 Jan 2017

Referee's report on:

**Physical theory for near-bed turbulent particle-suspension capacity**
**J.T. Eggenhuisen, M.J.B. Cartigny and J de Leeuw**

This contribution reconsiders the flow and sediment dynamics within a fully developed turbulent boundary layer and proposes a new dimensionless measure to determine the balance between the submerged gravitational weight of the suspension and the turbulent uplift. Some comparisons are given with experimental data.

In general this contribution raises some interesting issues but it requires careful revision to address several issues in their analysis.

1. Much of the analysis is based upon hydraulically smooth data sets and their interpretation, a regime corresponding to $u_* d/\nu \ll 1$. In this setting it is inevitable that the kinematic viscosity, $\nu$, plays a vital role in the mechanics and associated dimensional reasoning. However, how relevant are hydraulically smooth conditions to natural settings?

2. The key new empiricism is to write a quadratic function for the dimensionless acceleration (eqn 7) and hence a cubic equation for $\overline{w'^2}$. This function is determined by fitting data from DeGraaff and Eaton (2000). Are there theoretical grounds for expecting a quadratic dependence? Or rather is the important insight to elucidate the dimensional dependence of the acceleration? Also given the comments in (i), how relevant is this fitted form for natural systems with non-vanishing hydraulic roughness. Could it be that roughness enters the expressions if the boundary is not smooth?

3. I wonder whether it is helpful to describe the dynamics in terms of an 'acceleration'. Instead, presumably, the mean pressure field is altered by non-vanishing gradients of $\overline{w'^2}$ in order to satisfy the 'vertical' momentum balance and it is through this pressure field that the particle-scale dynamics are affected.

4. Following (iii), I think more must be said to justify the particle motion and to explain why it is appropriate that the particles follow the motion of fluid elements. My approach would be to form the mean pressure field (see (iii)) and deduce the stresses on the particle due to it. It is presumably necessary to average over the particle size and therein it is necessary to assume that that the particle diameter $d$ is much less than the flow lengthscale $\nu/u_*$, a condition that reduces to $u_* d/\nu \ll 1$. Furthermore a more thorough analysis of the particle motion would naturally identify the Stokes number as the important measure of the effects of particle inertia.

5. Much is made of the realisation that $\Gamma$ is independent of grain size (and/or settling velocity). For incipient motion in hydraulically smooth conditions, I suspect that this is already captured by the usual dependence of the critical Shields parameter $\theta_c \equiv u_*^2/(Rgd)$ upon the particle Reynolds number, $Re = u_* d/\nu$. When $Re \ll 1$, $\theta_c = K/Re$, where $K$ is a constant. Thus incipient motion is determined by $\theta_c Re = u_*^3/(Rg\nu) = K$. This conclusion is identical to what is derived in the paper, but is surely in line with a 'competence'-approach to modelling sediment transport.

6. The need for a reference concentration or a flux boundary condition in gradient diffusion models of sediment transport: this is of course, a long standing issue in sediment transport research and one for which steady flows do not shed much insight. For steady flows, one might prescribe a reference concentration at a small elevation above the bed. Alternatively, one might prescribe an erosive flux, but then since the suspension is in a steady balance between erosion and settling, this also leads to a prescribed concentration at the base. It is only for unsteady dynamics that the two types of boundary conditions behave differently - and I wonder what this new formulation can say about the concentration, or its flux, in this scenario?

7. It is assumed that the concentrations are sufficiently dilute so that the particles do not affect the flow and thus the clear fluid correlations of eqn (7) are applicable. However the theory is used for volumetric concentrations in excess of 0.1 and so I wonder how secure this assumption is? It is also used qualitatively to describe the collapse of the turbulence, which might well be a phenomena associated with high concentration suspensions.

Other minor comments follow:

1. Page 1, line 22: Be more precise with the proximity to the boundary.

2. Page 2, line 18: 'wall bounded'

3. Page 3, line 9: 'prime' rather than 'apostrophe'.

4. Page 3. $u_*$ ought to be defined in terms of the boundary shear stress.

5. Page 6. Better to say 'net gravitation pull'.

6. Page 8. The parameter $Z$ depends on grain size and so it is incorrect to say that the expression (14) is independent of grain size.

7. Page 10, line 21. I am not sure how the data from Cantero et al. is used because the the parameters are not identical. It must depend upon $w_s$.

8. References. There are many typos in the references - see, for example, the entries for Bagnold, Dorrell, Montes Vudela, Spalart and Vanoni.

---

## Author Response (AR2)

Associate Editor Decision: Publish subject to minor revisions (review by Editor) (14 Mar 2017) by Prof. Daniel Parsons Comments to the Author:
The authors have made a number of modifications based on the comments of the reviewers. This has tightened the paper. There do however remain some areas highlighted by the reviewers that the authors have not yet adequately dealt with in my view (A-D). I judge these to be technical in nature and not too taxing - thus I am satisfied that this does not need to be further-reviewed beyond me as AE. I ask for a further set of changes in response to A-D below and a point-by-point list of changed made to incorporate these changes into the next submission version.

A) CLEAR WATER CONDITIONS INDICATE THE SCALE OF FTURB. (Point 7, Reviewer 2 also)  > Can you add a few sentences and beef up the "justification" to satisfy this point more fully please - several of the reviewers raise this point and it is not really dealt with in the revisions. It is a basic assumption that is likely not valid - so you need to address this please.

> *We have emphasized this aspect in the following locations:*
> *P5L4-5*
> *P5L14-L33*

B) TURBULENCE EXTINCTION OCCURRS AT LOW AND HIGH ABSOLUTE CONCENTRATIONS  > Can you add a sentence on this in the introduction also - there are a few papers you could cite there in a background review that would make this section easier to manage and introduce.

> *We have added this insight as a main result of the paper in the introduction, with references to the supporting DNS and experiments (P1L37-P2L3).*

 C) Referee 2 Pt 2 - QUADRATIC
 his is an important point that is not adequately dealt with yet. Please revisit this.

> *The associate editor has urged us to consider this point again. This prompted us to review an authoritive textbook on the subject of turbulent scaling: "Turbulent Flows." By Pope [2000]. It turns out that this text does contain a fundamental theoretical consideration for the leading order dependency of $w^+$ on $z^+$. However, application of the theory does not lead to the exponential scaling hypothesized by the two referees. Moreover, comparison between the theory and the benchmark data is non-trivial. There is significant misfit between the theory of Pope, and the benchmark data of our Fig. 2. Main reason for this difficulty is that the proposed theoretical scaling (Eq. 7.61 in Pope, 200) depends on only the first term in a Taylor expansion of the spatial field of the Reynolds stresses. Many more terms would need to be incorporated in the theory to approach the benchmark, but this would lead to mathematical considerations of Reynolds-stress fields that are beyond our contribution. We are convinced that the wholly empiric approach we have followed in the assumption of the form of Eq. 7 leads to a correct numeric approximation of the distribution of $w^+(z^+)$, as evidenced by the success of Fig. 2.*
> *We have added explicit sentences describing the pragmatic empiricism of our Eq. 7 (P4L5-7 & P4L15-17).*

 D) Referee 2 PT 5 - GRAIN SIZE
 I do not agree that this is beyond the scope. Please revise the paper to include points pertaining to this advance - sight of the discussion around this is sufficient rather than additional results - but this is novel and will make the paper stronger.

> *We have now included this insight in the main text of the particle size discussion (P7L31-P8L4); and we have acknowledged the anonymous reviewer as a source in the Acknowledgment section (P13L8-10).*